# Hepatocellular Carcinoma: Old and Emerging Therapeutic Targets

**DOI:** 10.3390/cancers16050901

**Published:** 2024-02-23

**Authors:** Greta Pessino, Claudia Scotti, Maristella Maggi

**Affiliations:** Unit of Immunology and General Pathology, Department of Molecular Medicine, University of Pavia, 27100 Pavia, Italy

**Keywords:** liver cancer, HCC, immunotherapy, targeted therapies, tumor-associated antigens (TAAs), CAR-T, antibodies, targets, cancer-related pathways

## Abstract

**Simple Summary:**

Liver cancer is one of the most difficult solid tumors to treat and is responsible for one-third of cancer-related deaths worldwide. In particular, the quest for effective therapeutic strategies for hepatocellular carcinoma, which often arises from a chronic inflammatory background, remains an open challenge. In this review, we aim to provide an overview of the current therapeutic options available, focusing on recent advances in targeted therapies and the pursuit of emerging potential targets.

**Abstract:**

Liver cancer, predominantly hepatocellular carcinoma (HCC), globally ranks sixth in incidence and third in cancer-related deaths. HCC risk factors include non-viral hepatitis, alcohol abuse, environmental exposures, and genetic factors. No specific genetic alterations are unequivocally linked to HCC tumorigenesis. Current standard therapies include surgical options, systemic chemotherapy, and kinase inhibitors, like sorafenib and regorafenib. Immunotherapy, targeting immune checkpoints, represents a promising avenue. FDA-approved checkpoint inhibitors, such as atezolizumab and pembrolizumab, show efficacy, and combination therapies enhance clinical responses. Despite this, the treatment of hepatocellular carcinoma (HCC) remains a challenge, as the complex tumor ecosystem and the immunosuppressive microenvironment associated with it hamper the efficacy of the available therapeutic approaches. This review explores current and advanced approaches to treat HCC, considering both known and new potential targets, especially derived from proteomic analysis, which is today considered as the most promising approach. Exploring novel strategies, this review discusses antibody drug conjugates (ADCs), chimeric antigen receptor T-cell therapy (CAR-T), and engineered antibodies. It then reports a systematic analysis of the main ligand/receptor pairs and molecular pathways reported to be overexpressed in tumor cells, highlighting their potential and limitations. Finally, it discusses TGFβ, one of the most promising targets of the HCC microenvironment.

## 1. Introduction

According to the 2020 global cancer rating, liver cancer is the 6th most common cancer worldwide in incidence (https://www.wcrf.org/cancer-trends/liver-cancer-statistics/, accessed on 31 January 2024) and third in total cancer-related deaths (https://gco.iarc.fr/today/, accessed on 31 January 2024).

Histologically, two main subtypes of primary liver cancer can be distinguished, the most common subtype being hepatocellular carcinoma (HCC), with a 75–80% frequency, and the second most common being intrahepatic cholangiocarcinoma with a 10–15% frequency [1].

HCC has a strong association with chronic viral hepatitis, caused by hepatitis B and C virus infections (HBV and HCV, respectively) [2]. Other risk factors for HCC onset are non-viral chronic hepatitis (e.g., alpha-1-antitrypsin deficiency, hereditary hemochromatosis, and glycogen storage disease); alcohol-abuse-related steatosis; nonalcoholic steatohepatitis (NASH); environmental exposure to aflatoxin B1; and cigarette smoking [3]. No specific genetic alterations have been unequivocally associated with liver cancer tumorigenesis [4].

Therapeutically, HCC represents one of the most complex solid tumors to treat, with surgery and liver transplantation remaining the most effective treatment options [5,6]. Conventional chemotherapy is reserved for patients with the most advanced disease [5,7], and a particularly hostile tumor microenvironment makes it difficult for immunotherapy to achieve the intended results [8,9]. Therefore, the search for alternative, targeted therapeutic approaches is of the highest priority.

The aim of this review is to provide a comprehensive overview of the therapeutic targets for the treatment of HCC, from the best known to the emerging ones, focusing on both open issues and future perspectives (Figure 1).

## 2. Targeted Therapy: State of the Art

### 2.1. General Standard of Care

The strategy for liver cancer therapy depends on many factors, including the histology, size of the lesion(s), their multiplicity, and stage of the disease. Comprehensive reviews about the main therapies adopted report surgical resection, liver transplantation, and loco-regional treatments as the main surgical and physical approaches [5,10]. Systemic chemotherapy is reserved for advanced HCC and includes doxorubicin, gemcitabine, and oxaliplatin [5]. The resulting overall survival is, however, still low, and alternative systemic therapies are under evaluation. Sorafenib (brand name: Nexavar®) and regorafenib (Stivarga®) are kinase inhibitors used to treat unresectable liver carcinoma (https://go.drugbank.com, accessed on 25 January 2024). Sorafenib is a bi-aryl urea and an oral multikinase inhibitor. It targets cell surface tyrosine kinase receptors and downstream intracellular kinases that are implicated in tumor cell proliferation and tumor angiogenesis [11]. This mechanism of action is shared by regorafenib: in in vitro biochemical or cellular assays, regorafenib or its major human active metabolites, M-2 and M-5, inhibited the activities of RET, VEGFR1, VEGFR2, VEGFR3, KIT, PDGFRα, PDGFRβ, FGFR1, FGFR2, TIE2, DDR2, TrkA, Eph2A, RAF1, BRAF, BRAFV600E, SAPK2, PTK5, and Abl at concentrations of regorafenib that have been achieved clinically (https://go.drugbank.com, accessed on 25 January 2024). In in vivo models, regorafenib demonstrated anti-angiogenic activity in a rat tumor model and inhibition of tumor growth, as well as anti-metastatic activity in several mouse xenograft models, including some for human colorectal carcinoma. Improvement in survival, though statistically significant, is, however, very limited with both drugs [5,12,13,14,15,16], which still imposes the need for better therapies.

### 2.2. Immunotherapy

Cancer cells are able to block immune cell activity by exposing on their surface the programmed cell death ligand (PDL-1, CD274) that interacts with receptors, called immune checkpoints, present on the surface of T-cells, like the B- and T-lymphocyte attenuator (BTLA), the programmed death protein 1 (PD-1), and the cytotoxic T-lymphocyte-associated protein 4 (CTLA4) [17]. The resulting signals block T-cell-mediated immune responses to neoplasms and T-cell activation, allowing cancer cells to escape scrutiny and grow uncontrollably. Immune checkpoint inhibitors are drugs able to modify the tumor microenvironment, turning inactive immune cells into cells able to actively fight the tumor itself. There are currently six FDA-approved checkpoint inhibitor immunotherapies for liver cancer, for which the main features are summarized in Table 1. All of them are antibodies. Two target PD-L1, atezolizumab (Tecentriq™) and durvalumab (Imfinzi™), two are anti-PD1 antibodies, pembrolizumab (Keytruda®) and nivolumab (Opdivo™), and two are anti-CTLA4 antibodies, ipilimumab (Yervoy™) and tremelimumab (Imjudo®). By interfering with the immune checkpoints, these molecules can remove the inhibition induced by the tumor from T-cells and allow the activation of the latter, unleashing their cytotoxic activity. Combinations of two immune checkpoint inhibitors are now known to enhance clinical responses, and the administration of atezolizumab along with bevacizumab has become the new standard of care for first-line unresectable HCC (Table 1).

Although the field of classical, full-length monoclonal antibodies against HCC is thriving, alternative antibody-engineered solutions are still at the experimental stage and have had limited development. As further discussed in a following section, an anti-FGF2 diabody was found to inhibit the expression of PD-L1 and the epithelial mesenchymal transition (EMT) of hepatoma cells, suggesting a potential clinical application in inhibiting metastasis and immune escape [22], while a bispecific single-chain diabody targeting c-Met and PD-1 substantially decreased the tumor burden in an HCC model of high c-Met expression [23], which has led to the development of an analog, CAR-T [24].

### 2.3. Antibody–Drug Conjugates (ADCs)

Antibody–drug conjugates (ADCs) are complex molecules consisting of a targeting component, represented by an antibody or an engineered antibody fragment, joined to a toxic payload by a labile linker. The linker can be cleaved inside the cancer cells, after ADC internalization, typically at the level of the lysosomes. There are not yet ADCs approved for the treatment of HCC, though there is experimental work supporting their potential [25]. In this respect, two targets have been explored so far: the cell surface membrane-bound glypican-3 (GPC3), a heparan sulfate proteoglycan protein, and the cell membrane-bound CD24, which is a mucin-like molecule that is overexpressed in a range of human carcinomas, including HCC [26].

In the case of GPC3, two humanized mAbs (hYP7 and hYP9.1b) in the IgG format have been reported to induce antibody-dependent cell-mediated cytotoxicity (ADCC) and complement-dependent cytotoxicity (CDC) in GPC3-positive cancer cells [27]. Two ADCs composed of hYP7 coupled with duocarmycin or pyrrolobenzodiazepine dimer (PC) were able to kill cancer cells, with the latter being 5–10 times more potent than the former. In fact, a single treatment of hYP7-PC induced tumor regression in multiple mouse models [28].

By connecting a CD24-targeting antibody (G7mAb) to two doxorubicin molecules [29] or monomethyl auristatin E [30], the resulting ADCs were able to suppress tumor growth, decrease systemic toxicity, and prolong the survival of HCC-bearing nude mice.

Given the success of ADCs, like Adcetris™ (brentuximab vedotin), approved in 2011 for Hodgkin’s lymphoma, and Kadcyla™ (trastuzumab emtansine or T-DM1) approved for breast cancer in 2013, it is likely that this drug category might reserve positive surprises in the medium-long term in the treatment of HCC as well. 

### 2.4. Chimeric Antigen Receptor T-Cells (CAR-Ts)

Following its success in hematology since its FDA approval in 2017 [31], cellular immunotherapy with chimeric antigen receptor T-cells (CAR-Ts) has been put to the test for several solid tumors, including HCC. A review of clinical trials relevant to the field was recently published [32]. T- and NK cells transduced with a CAR that recognizes the surface marker CD147 (basigin) could effectively kill various malignant HCC cell lines in vitro and HCC tumors in xenograft and patient-derived xenograft mouse models [33]. In fact, CD147 is over-expressed in several types of cancer, including HCC [34,35], and plays an important role in the tumor microenvironment’s (TME’s) regulation, particularly through the production of proteases involved in cell matrix degradation, which promotes cell invasion and metastasis [35,36].

CD133 is a marker that has long attracted interest as a potential therapeutic target, being expressed by cancer stem cells (CSCs). A Phase II study investigated CD133-directed CAR-T cells in adults with HCC, obtaining, out of 21 evaluable patients, 1 partial response, 14 stable diseases from 2 to 16.3 months, and 6 progressions [37].

CAR-Ts targeting GPCR3 were successful in suppressing tumor cell growth in HCC patient-derived xenografts [38]. GPC3 is a cell-membrane-anchored HSPG proteoglycan for which the overexpression in HCC cancer cells correlates with a poor prognosis [39] and that exerts its pro-tumoral function via the Wnt-signaling pathway [40]. A phase I trial established the initial safety profile along with early signs of the antitumor activity of GPC3-directed CART-s in patients with advanced HCC [41]. The potentiation of this effect was observed, including CD28, 4-1BB, or both, in two xenogeneic tumor models [42]. A clinical trial is currently underway to test the safety of a dual CAR-T directed against both CD147 and GPCR3 in patients with advanced HCC (NCT03993743). CAR-T cells engineered to co-express IL-7 to induce proliferation and CCL19 to promote CAR-T migration to the tumor site have shown some of the most encouraging effects. The administration of this type of cells to patients with advanced GPC3+ HCC has yielded astonishing results in terms of intratumoral activity, with one patient having all the tumor cells eliminated 30 days after administration (NCT03198546) [43].

An anti-alpha-fetoprotein (AFP) CAR-T has shown potent anti-tumor activity in vivo in mouse models [44], and a Phase I clinical trial has been conducted in patients with AFP+ HCC, the results of which are not yet available (NCT03349255). Alpha-fetoprotein is, in fact, an established marker in HCC, being overexpressed in the vast majority of patients and being associated with drug resistance and increased tumor progression [45,46].

HBV-related advanced HCC has been treated with HBV-specific TCR expressed on T-cells, showing to be well tolerated, with most of the patients exhibiting a reduction in or stabilization of circulating HBsAg and HBV DNA levels [47]. Short sequences of integrated HBV-DNA present in HCC cells have also been found to encode epitopes that are recognized by and activate engineered T-cells, with initial clinical response [48].

Finally, NKG2DLs (NK group 2 member D ligands) are overexpressed in HCC cancer cells but absent in healthy ones [49], representing a promising target. In this sense, an example is MICA/B, an activating receptor that belongs to the class of NKG2DLs and is thus able to activate the cytotoxic effect of effector cells that express NKG2D, such as NK cells and CD8+ cells. The targeting of MICA/B has given promising results in vitro in patients with cholangiocarcinoma (CCA) [50,51].

NKG2D-expressing CAR-Ts were, therefore, designed to direct their action against tumor cells overexpressing NKG2DL, with good results obtained both in vitro and in vivo. NKG2D-based CAR-T cells comprising the extracellular domain of human NKG2D, 4-1BB, and CD3ζ-signaling domains (BBz) effectively eradicated SMMC-7721 HCC xenografts [49]. A clinical trial is underway to evaluate the safety of the administration of this CAR-T to patients with HCC (NCT04550663).

CD155 is another emerging tumor target. Initially identified as a poliovirus receptor (PVR) [52], it is now known to be an immunoglobulin superfamily adhesion molecule involved in many different physiological processes ranging from cell adhesion and migration to the proliferation and modulation of immune responses [53]. It is also overexpressed in tumors, including HCC, interacts with the activating receptor DNAM-1 expressed on NK cells, and is involved in anti-tumor immune responses [54]. CD155 is also a high-affinity ligand of TIGIT [55], and interfering with the TIGIT-CD155 axis has shown promising results in enhancing CAR-T efficacy in experimental models [56].

## 3. Review of Established and Novel Potential Targets in HCC

HCC is acknowledged as a complex tumor, particularly in its tumorigenesis, stemming from a multi-step process involving adaptive genetic and epigenetic alterations [57,58]. In the last decade, efforts to better understand the cancer biology of this challenging tumor have revealed recurrent alterations in foundational signaling pathways within cell biology, such as tyrosine receptor pathways, RAS/MAPK, JAK/STAT, PIK3/AKT/mTOR, WNT/β-catenin, and cell cycle regulation [59,60,61,62,63]. This ignited the search for suitable targets for targeted therapy, which has proven to be, and remains, particularly challenging, as these pathways are broadly active in most cells. However, a progressively deeper understanding of the individual actors causing such altered regulation has provided a more specific focus on molecular and cellular components that could result in effective targets. In the following section of this review, we report the results of a survey performed through an extensive literature search (Table 2 and Table 3). Of particular interest in this regard is the proteomic work performed in [64], which also revealed as-of-yet unclassified molecules potentially useful as therapeutic targets.

Given the broad nature of the topic covered, the following discussion has been structured first to cover the receptors and ligands primarily involved in the altered signaling found in HCC and then to focus on the major affected pathways.

The hope is that this work will provide a fresh look at the best-known targets of HCC and new perspectives for the search for novel therapeutic targets.

### 3.1. Receptor Tyrosine Kinases (RTKs)

Alterations in the signaling pathways mediated by growth factors (GFs) and receptor tyrosine kinases (RTKs) are known to have a leading role in oncogenesis. GFs are polypeptides carrying mitogenic messages and are considered as mediators of paracrine cellular communication, acting mostly on cells located near the one that secreted them. With more than 100 GFs and 58 RTKs grouped in 20 families, these pathways are involved in the regulation of numerous and variable cellular processes [65]. All the RTKs are activated through a unique mechanism, consisting of dimerization followed by the autophosphorylation of tyrosine residues located near the C-terminal of the intracellular domain. These phosphotyrosine residues can be recognized by proteins carrying the so-called SH2 (SRC homology domain 2) and PTB (phosphotyrosine-binding domain) domains, which can implement the signal either by being phosphorylated (e.g., JAK) or by being recruited as docking proteins, starting the formation of protein complexes, which then initiate larger signaling cascades (e.g., RAS/MAPK and PI3K pathways).

RTK signaling is known to be severely altered in HCC [58], and, along with their soluble ligands, RTKs have a crucial role in redirecting the crosstalk occurring in the TME toward pro-tumoral functions.

In the following section, we will briefly discuss the main classes of RTKs, along with their ligands, involved in oncogenesis and the progression of HCC and their potential as therapeutic targets.

#### 3.1.1. Epidermal Growth Factor Receptors (EGFRs)

The erythroblastic leukemia viral oncogene homolog (ErBb) family consists of the epidermal growth factor receptors, EGFRs, also called ERBB1 or HER1, ERBB2, ERBB3, and ERBB4, and plays a pivotal role in the progression of various cancers. These receptors are implicated in the regulation of fundamental cellular processes, including proliferation, regeneration, survival, migration, angiogenesis, tumorigenesis, and metastasis [66].

Within this family, the class I EGFR is a transmembrane receptor composed of an extracellular domain to which ligands bind, followed by a transmembrane domain and an intracellular domain, where the tyrosine kinase domain and the carboxy-terminal tail containing key tyrosine residues are located [67,68]. Upon ligand binding, homo- or heterodimers are formed with the related receptors, ERBB2, ERBB3, and ERBB4, and kinase activation initiates multiple intracellular signaling pathways.

EGFR can be activated by several ligands, like the epidermal growth factor (EGF), the transforming growth factor α (TGFα), amphiregulin (AR), epiregulin (EREG), betacellulin (BTC), heparin-binding EGF (HB-EGF), and epigen (EPGN) [69,70]. The subsequent signaling events have been summarized in several classical reviews [68,71]. 

Studies have revealed a significant association between EGFR and liver cancer. EGFR is overexpressed in human cirrhotic liver tissue and in 66% of human HCCs, correlating with aggressive tumors, metastasis, and poor patient survival [72]. Aberrant EGFR activation through gene amplification and/or mutation, though common in various types of cancer, is not frequently found in HCC [66]. Notably, EGFR is detected in endothelial cells within cancerous tissue but is absent in healthy liver tissue, hinting at a potential avenue for therapeutic selectivity [73]. 

EGFR is expressed in liver macrophages both in human HCC and mouse HCC models, being critical for HCC development, as demonstrated by specific deletion in KCs/macrophages [74]. This points to the role of EGFR in the interaction between the components of the TME. Recently, it has also been shown that EGF/EGFR can mediate EMT through the AKT/glycogen synthase kinase-3β (GSK-3β)/snail signaling pathway to promote HepG2 cell proliferation and migration [75]. The pharmacological or genetic inhibition of EGFR in different models of hepatic injury ending in HCC prevents tumor development [76,77,78].

Because of all these observations, EGFR represents, in principle, a rational target for anti-tumor strategies. However previous attempts with anti-EGFR agents have shown no effective response in HCC patients [79,80]. These observations are ascribed to increasing evidence of EGRF acting as a “signaling hub” where different extracellular growth and survival signals converge [81] thanks to transactivation, the process whereby a primary agonist, via binding to its receptor, activates a receptor for another ligand. This situation can typically occur with G-protein-coupled receptors (GPCRs), leading to the activation of growth factor receptors. For example, the binding of an agonist to a GPCR in primary cells leads to the release of epidermal growth factor (EGF), which in turn activates EGFR [82].

Both ligand-dependent and ligand-independent transactivations of EGFR have been described [81], pointing to the high complexity of this signaling system. 

Even regulators of EGFR signaling have been described as altered, e.g., ERBB receptor feedback inhibitor 1 (ERRFI1), which is deleted in HCC [83].

Another protein of interest interacting with EGFR is the urokinase-type plasminogen activator receptor (uPAR). Many studies have shown increased levels of the urokinase-type plasminogen activator receptor (uPAR) in HCC, which are related to liver metastasis and a poor prognosis [84,85,86,87]. The urokinase-type plasminogen activator is a key protein in the plasminogen activation system, which plays a proteolytically important role in the invasion and metastasis of various cancer cells. Attempts to target it in other cancer types is leading to interesting results, even in the surgical domain [88], and it is, therefore, not surprising that it has now become the focus of interest for integrated antitumor therapeutic strategies [87]. 

In addition to EGFR, ligands, such as TGFα, EGF, HB-EGF, AR, and BTC, along with ADAM17, show heightened expressions in human liver tumor cells and tissues [70]. EGFR ligands are also overexpressed in chronic liver injury, as shown both in experimental models and human cirrhotic tissue [89]. This may favor the hepatocarcinogenic process.

TGFα, a cytokine known for its mitogenic effect on hepatocytes, has been implicated in promoting liver carcinogenesis [90]. Intriguingly, it was suggested that the upregulations of TGFα and EGFR in hepatocytes, particularly in cirrhotic hepatocytes, are induced by a state of hypoxia, supporting their therapeutic potential in liver cancer [91]. The role of EGF, however, remains less clear in HCC progression. Evidence suggests its involvement in creating an autocrine loop leading to increased expression of TGFβ, thereby favoring HCC progression [92]. This intricate interplay of ligands and receptors underscores the complexity of the molecular landscape in liver cancer.

Beyond EGFR, recent findings highlight the overexpressions of ERBB2 and ERBB3 in different tumor cohorts [93]. ERBB3, in particular, has been linked to lapatinib sensitivity induced by the HBV-X protein [94].

The ERBB family members (ERBB2, ERBB3, and ERBB4) are also the dominant receptors for neuregulin 1 (NRG1), which has been identified as a positive regulator of the HCC EMT and metastasis [95]. This further underscores the multi-faceted role of ERBB receptors in shaping the aggressive phenotype of HCC.

#### 3.1.2. Platelet-Derived Growth Factor Receptors (PDGFRs)

Platelet-derived growth factor receptors (PDGFRs) are class III tyrosine kinase receptors localized in the plasma membrane in the form of α or β monomers. PDGFs and their ligands are divided into four classes (A, B, C, and D), which can pair to form five different dimeric configurations (AA, AB, BB, CC, and DD) [96,97,98,99,100,101,102].

Binding to the respective ligands leads to the formation of dimers that can be composed of PDGFR-αα, -αβ, and -ββ. PDGFRα and PDGFRβ share different ligands and downstream effectors, and their distinct functions at a regulatory level in different cell types are mainly due to differences in spatial–temporal expression patterns [103]. PDGFs are mainly released by endothelial and epithelial cells, macrophages, and platelets after degranulation [104]. 

The PDGF/PDGFR axis is implicated in the regulation of several signaling pathways involved in cell growth and differentiation, particularly of mesenchymal stem cells [105]. Major ones include the RAS/MAPK, PI3K-AKT, and PLC gamma pathways [106].

The altered expression levels and upregulation of the PDGF/PDGFR-signaling axis have been found in several types of cancer [104,107]. These can occur at the level of the tumor cells themselves through autocrine signaling [108] or at the level of non-tumor cells in the TME [109,110]. In the liver, PDGFs are important proliferative agents, particularly for hepatic stellate cells (HSCs), which are an important cellular component in liver fibrosis and cirrhosis [111,112], as well as in the TME of HCC. Here, HSCs play an important role in facilitating tumor progression by actively contributing to the processes of angiogenesis, acquisition of invasive capabilities, and metastasis [113], as well as in promoting an immunosuppressive environment through the induction of T-regs (regulatory T-cells) and MDSCs (myeloid-derived suppressor cells) [114,115].

It was previously demonstrated that PDGFRα overexpression in HCC is a prognostic marker [116], while PDGFRB showed the highest overexpression rates among the potential HCC drug targets [93]. However, the role of PDGFRs in HCC remains complex, with research suggesting its involvement in various aspects of the disease.

In particular, the role of PDGFRα in HCC carcinogenesis appears to be of increasing importance. There is growing evidence linking altered PDGFRα-mediated signaling with increased tumor progression and metastasis [117,118]. In particular, this would be due to not only its critical role in promoting angiogenesis during HCC development but also its subsequent metastatic progression [119,120].

In addition, the strong involvement of the PDGFA/PDGFRα axis in promoting the epithelial–mesenchymal transition (EMT) of HCC, as confirmed by several in vivo studies [121,122,123,124], further supports the correlation between its altered signaling and tumor metastatic capacity. A more detailed analysis of the mechanisms by which PDGFRα is involved in promoting tumor progression in HCC can be found in the comprehensive review by Kikuchi et al. [125].

Targeting strategies for the PDGF/PDGFR axis essentially involve the use of neutralizing antibodies directed against the receptor/ligand or small-molecule inhibitors [104]. There are currently no reports regarding the development of CAR-T directed against these receptors. To date, only one monoclonal antibody directed against PDGFRα is approved for the treatment of tissue sarcoma (olaratumab), while a large number of small-molecule inhibitors directed against PDGRFα are used for the treatment of several cancers, including sunitinib (renal cell carcinoma), imatinib (leukemias and myelodysplastic/myeloproliferative disease), ripretinib (gastrointestinal stromal tumors), and erdafitinib (urothelial carcinoma) (https://go.drugbank.com, accessed on 25 January 2024). In HCC in particular, the importance for targeting these receptors is demonstrated by the fact that sorafenib and regorafenib, among the main first-line drugs in the treatment of unresectable HCC, are multi-tyrosine kinase inhibitors that, among others, precisely act against PDGFRs [12,126].

#### 3.1.3. Fibroblast Growth Factor Receptors (FGFRs)

As for class IV receptors, an alteration in the fibroblast growth factor (FGF)/fibroblast growth factor receptor (FGFR) signaling axis, which regulates cell proliferation, differentiation, and survival, is observed in multiple cancers, including HCC [127]. In fact, the levels of at least one FGF8 subfamily member and/or one FGFR are upregulated in 82% of HCC cases [128].

On the receptor side, FGFR3 and FGFR4 are the major FGFRs overexpressed in HCC, while the upregulations of FGFR1 and FGFR2 are rarely observed. FGFR3 isoforms have already been proposed as novel therapeutic targets years ago [129,130]. FGFR4 has recently been identified as an oncogenic-driver pathway for HCC patients [131], as also confirmed by proteomic analysis [93].

Regarding the soluble ligands, FGF2, which is barely detected in nonparenchymal cells or noncancerous liver tissue, is overexpressed in HCC, like FGF8, FGF17, and FGF18. Moreover, FGF19 plays a crucial role in liver carcinogenesis and progression.

Components of the FGF1 subfamily act through autocrine signaling and have long been implicated in promoting proliferation, invasiveness, and angiogenesis in HCC. Notably, FGF1 and FGF2 are expressed in the presence of chronic disease in the liver, and their elevated expression levels correlate with more advanced tumor stages [132,133,134,135]. Of the FGFs, FGF2 has been the most extensively studied, and FGFR3 appears to be most involved in HCC progression [136,137,138]. As previously discussed, an anti-FGF2 diabody could represent a potential therapeutic agent for the inhibition of metastasis formation and immune escape because it was able to inhibit the anti-PD-L1 expression and EMT in hepatoma cells [22]. 

In the FGF8 subfamily, FGF8, FGF17, and FGF18 components bind to FGFR2, FGFR3, and FGFR4 receptors, respectively. They act as mediators of paracrine signaling and have been found to be actively involved in the promotion of polypharylation, malignancy, and angiogenesis in HCC [128]. Notably, FGF8 is involved in conferring resistance to the EGFR inhibitors [139].

On the other hand, the FGF19–GFR4–KLB signaling pathway is recognized as a major driver of disease initiation and progression, being involved in the regulation of cell proliferation, survival, EMT, migration, and invasion in HCC cells [132,140,141,142,143,144,145,146,147]. 

Components of the FGF19 subfamily act as endocrine hormones, of which FGF19 is the main representative involved in liver cancer.

FGF19 does not appear to be expressed in the liver under physiological conditions, whereas its elevated presence has been reported in several liver diseases (hepatitis C, cirrhosis, and HCC) [132,148]. The overexpression of FGF19 at the protein level in HCC tissues is also closely correlated with larger tumor sizes, advanced disease stages, and early recurrence [132,149,150]. 

Given the important role of this receptor family in the development and progression of HCC, several inhibitors of the FGF/FGFR signaling pathway have been developed over the years, from the early multi-target pan-FGFR to the more recent selective FGFR4 inhibitors currently in Phase I/II clinical trials (NCT04194801 and NCT02508467, respectively).

#### 3.1.4. Vascular Endothelial Growth Factor Receptors (VEGFRs)

Vascular endothelial growth factors (VEGFs—VEGFA, VEGFB, VEGFD, VEGFE, and PIGF) are an important class of growth factors playing a critical role in the regulation of angiogenesis and vascular permeability through interaction with their receptor VEGFRs (VEGFR1, VEGFR2, VEGFR3, and class V RTKs) [151]. Specifically, VEGFR1 is more characteristic under pathological conditions, including inflammation and cancer, whereas VEGFR2 signaling appears to be more focused on the regulation of vascular endothelial cell proliferation and permeabilization, as well as survival signaling pathways [151,152]. The regulation of many factors involved in this signaling axis is mediated by the hypoxia-inducible factor (HIF) [153,154].

The canonical VEGF-R1/R2-mediated signaling pathway results in the activation of several kinases that are involved in important pathways regulating cell proliferation, migration, and spreading, as well as angiogenesis and vascular permeability, as mentioned above [152]. Among the major ones, there are PLC-γ, PI3K, AKT, RAS, SRC, and MAPK [155].

The implication of the VEGF/VEGFRs axis in the development and progression of several solid tumors has long been recognized [156,157]. The overexpressions of VEGFs/VEGFRs by stromal cells present in the TME induce angiogenesis by stimulating endothelial cell proliferation and increased vascular permeability [158,159,160]. However, VEGFs/VEGFRs are found to be also overexpressed in cancer cells, suggesting autocrine or paracrine signaling that directly stimulates the growth of the cancer cells themselves [156,161]. 

There is evidence that the VEGF/VEGFR autocrine signal axis is present in HCC cancer cells, with the overexpression of both promoting their growth and progression [157,162,163,164,165]. Recently, a large set of virus-associated liver cancers was classified according to proteomic profiling, whole-genome sequencing, and transcriptional analysis [64]. Among the three proteomic subclasses defined (R1, R2, and R3), the R2 subclass contains advanced proliferative tumors with TP53 mutations, high expression of VEGF receptor 2, and the worst prognosis but also four known targets of multikinase inhibitors: VEGFR2, CRAF, BRAF, and c-Kit, where c-Kit represents an as-of-yet underexplored target. VEGFR2 was overexpressed in HCC classified as R2 (the most aggressive cancer type) in [64], while VEGFR1-HIF1A was found to be selectively upregulated in the R3 proteomic class. The latter class corresponded to smaller and less aggressive tumors enriched with CTNNB1 mutations and possessing elevated mTOR-signaling-pathway activity.

From a clinical point of view, the high expression of VEGF in the tumor tissue and serum of patients with HCC correlates closely with the metastatic stage of the tumor as well as its size [166,167,168]. All these data confirm the therapeutic relevance of this family of targets.

Several of the small-molecule multi-tyrosine kinase inhibitors previously discussed for PDGFRs also act on VEGFRs (e.g., sunitinib), and, in particular, sorafenib and regorafenib, for which their use is well established in HCC therapy. In addition, several antibodies directed against VEGF/VEGFRs have been developed for the treatment of HCC. The first to be approved for the treatment of many solid tumors was bevacizumab, an anti-VEGF ligand antibody, which is currently being investigated for its therapeutic efficacy in HCC in several Phase III clinical trials, particularly in combination with atezolizumab (NCT04487067, NCT04732286, NCT04102098, NCT03434379, and NCT05904886). Ramucirumab is an anti-VEGFR2 antibody that is being evaluated in Phase III clinical trials (NCT02435433 and NCT01140347), with positive results in terms of increasing the overall survival (OS) and progression-free survival (PFS). In recent years, many efforts have been made to develop CAR-Ts targeting the VEGFR family of receptors. At present, none has reached the stage of clinical development, but in both in vitro and in vivo experiments, it has been observed that the use of CAR-T cells targeting VEGFR1, VEGFR2, and VEGFR3 positively prolonged the survival of mice and inhibited the growth of several types of solid tumors [169,170,171,172], although none of these studies has been performed in HCC so far. A detailed review on this topic can be found in [173].

#### 3.1.5. Mesenchymal–Epithelial Transition Factor (c-Met)

Hepatocyte growth factor (HGF) is a growth factor that was initially identified for its mitogenic properties toward hepatocytes [174,175,176] and that, over the years, has been shown to have a complex role in the regulation of liver functions, in particular, in cell motility, angiogenesis, immune responses, cell differentiation, and anti-apoptotic effects [177]. In HCC, it is known that both stromal and tumor cells release HGF in the TME, which affects tumor cells through autocrine and paracrine signaling by binding to their c-Met receptors [162].

c-Met belongs to the tyrosine kinase receptor family and interacts with its high-affinity ligand, HGF, through four regions called hotspots [178]. We refer to HGF-mediated c-Met activation as the canonical pathway, which, through homodimerization, induces signaling pathways, such as RAS/MAPK, ERK, PI3K, p38, and AKT/PKB [179,180].

However, c-Met can also be activated through the so-called non-canonical pathways, which are mostly associated with tumor progression, metastasis, and drug resistance [174,177,181]. Some of the most important include activation by des-gamma carboxy prothrombin (DCP), which is highly secreted by HCC cells [182]; interaction with other receptors, such as EGFR, integrin, beta-catenin, CD44, MUC1, and FAK [177,183,184]; hypoxia and miRNA loss [185]; and receptor-activating mutations and amplification [177].

The role of c-Met in liver disease is dual, as its overexpression is generally necessary to activate hepatocyte proliferation and liver regeneration in case of injury and to slow the progression of injury by suppressing chronic inflammation and fibrosis progression. At the same time, however, its overexpression has been shown to be involved in promoting the initial development and later progression of HCC [186].

Although not all cases of HCC are characterized by c-Met overexpression, where it is found, it correlates with a worse prognosis [187]. In contrast, HGF expression is found to be decreased in HCC tissue, whereas it is increased in peritumoral tissue [177]. Nevertheless, several studies correlate the high activity of this axis with the invasiveness and metastatic capacity of HCC [188], with an important role played by the TME, where high levels of HGF secreted by neutrophils, mesenchymal cells, and CAFs appear to be critical for the acquisition of these invasive features [189,190,191].

Attempts have also been made to use HGF levels in the serum of HCC patients as a diagnostic tool, which resulted, however, in being inadequate when considered as a single indicator [192,193]. In fact, the difference between the serum HGF levels of patients with cirrhosis and patients with both cirrhosis and HCC was not significant, but with a diagnostic sensitivity of 90.62%, the potential of this indicator is evident [193]. 

It is, therefore, well established that the HGF/c-Met axis represents a valuable therapeutic target in HCC, with increasing evidence both in vitro and in vivo [194,195,196]. Over time, the need to focus on the interaction of c-Met and downstream signaling mediators rather than the classical HGF/c-Met interaction is becoming more and more evident, given the greater involvement of non-canonical activation pathways in the c-Met signaling pathway.

Currently, 10 inhibitors of c-Met have been tested in clinical trials [186], belonging to three macro categories: selective c-Met tyrosine kinase inhibitors (TKIs); multi-targeted TKIs, including c-Met; and monoclonal antibodies against HGF and c-Met [197]. However, no effective treatment for HCC based on traditional TKI monotherapy has been found. Particularly promising was tivantinib, a highly selective c-Met inhibitor, which was able to increase the median time to progression and OS in a Phase II study [198]. Unfortunately, the subsequent randomized Phase III trial showed no significant OS increase in patients with advanced HCC and high c-Met expression treated with sorafenib compared to a placebo (NCT01755767), compromising the possibility of further development for clinical use.

Other c-Met inhibitors are being evaluated in clinical trials for the treatment of HCC. These include the non-selective TKI inhibitor cabozantinib in Phase III (NCT01908426), which increased OS from 8 to 10.2 months compared to a placebo and is currently being evaluated in combination with atezolizumab versus sorafenib in another Phase III trial (NCT03755791). Among the selective inhibitors, tepotinib showed promising results in a Phase II trial versus sorafenib (NCT01988493), and capmatinib is being tested in Phase II trials (NCT01737827).

The major issues detected in early clinical trials with c-Met inhibitors are mainly related to severe side effects. A possible solution to this problem would be to target downstream effectors, among which several have been proposed, including the Grb SH2 domain, Src, MAPK, STAT3, and Shp2 [199], some of which are also discussed in this work. However, their involvement in pathways regulated by numerous other RTKs makes them risky to target, as the results of such treatments would be difficult to predict. Therefore, the identification of downstream effectors in the c-Met pathway that may be more specific remains an open question.

It is also important to mention the use of miRNAs as an alternative to inhibitors for c-Met targeting. This therapeutic approach would be based on non-coding RNA molecules that can regulate protein expression by suppressing mRNA translation [200], which is proving to be increasingly promising in preclinical studies [201,202,203,204,205]. In the clinic, however, the current studies are mainly evaluating the potential role of circulating miRNAs for diagnostic and prognostic purposes (NCT02928627, NCT05148572, and NCT02448056).

Concerning other promising therapeutic approaches, good results have been obtained by testing CAR-T directed against c-Met in different types of solid tumors, such as renal, gastric, and breast cancers [206,207,208,209]. In HCC, the best results were obtained with a dual CAR-T directed against both c-Met and PD-1, which showed a good improvement in anti-tumor activity along with the increased persistence of CAR-T cells after administration (NCT03672305) [24]. 

All in all, despite the complexities encountered in the development of effective targeting strategies for the HGF/c-Met axis, its therapeutic potential has been extensively demonstrated, confirming the need for further efforts to effectively target this important pathway.

### 3.2. Toll-Like Receptors

Toll-like receptors (TLRs) are a family of transmembrane receptors involved in innate immunity and located in the plasma membrane (TLR1, TLR2, TLR4, TLR5, TLR6, and TLR10) or in the endosomal membranes (TLR3, TLR7, TLR8, and TLR9). Their activation occurs through dimerization commonly induced by the recognition of PAMPs (pathogen-associated molecular patterns) or DAMPs (damage-associated molecular patterns) released owing to infection, tissue damage, or necrosis.

Several studies confirm that alterations in TLR signaling have a significant impact on HCC development and progression. Recently, a TLR-based gene signature able to act as a robust prognostic indicator in HCC was identified, further confirming their pivotal role [210]. In particular, TLR2 and TLR4 are known to orchestrate the so-called hepatic inflammation–fibrosis–carcinoma sequence (IFC), which frequently leads to HCC onset [211]. In fact, TLR2 and TLR4 activations trigger the signaling of NF-kβ and MAPK, which eventually leads to the release of several inflammatory mediators, such as TNFα and cyclooxygenase 2 (COX-2) [212,213,214,215]. 

TLR2 is found to be functionally expressed on the cell surface of almost all types of liver cells, including hepatocytes [216] and Kupffer, stellate, and sinusoidal endothelial cells [217]. Its role in the promotion of chronic liver disease progression is well assessed [218], while growing evidence in recent years has also been elucidating its involvement in the promotion of HCC development, making it a promising therapeutic target [219,220]. It has been shown in vitro and in vivo that the knockdown of TLR2 could inhibit cultured-HCC growth and proliferation [220]. It has also been demonstrated that TLR2 has an impact on HCV viral loads, leading to higher risk of HCC onset in HCV-infected patients [221]. Recently, a correlation between the expression of TLR2 and other markers of malignancy in HCC has been established, in particular with proliferation, apoptosis, and vascularization HCC markers (Ki-67, Caspase 3, and VEGF, respectively) [222]. For the first time, a vaccine based on a TLR2 ligand is being tested in combination with atezolizumab for the treatment of HCC in a Phase I clinical trial (NCT05937295).

TLR4 has also been intensively studied for its role in carcinogenesis, metastasis, and cancer progression [223,224]. In the liver, TLR4 is normally expressed on the cell surface of hepatocytes, Kupffer cells, and hepatic stellate cells but at relatively low levels. However, in the presence of liver damage, TLR4 expression is upregulated [225]. Growing evidence supports TLR4’s crucial role in the so-called liver inflammation–fibrosis–carcinoma (IFC) sequence, modulating the inflammatory response in viral hepatitis, autoimmune liver disease, and alcohol-induced liver damage. It is well documented that TLR4 expression is increased in livers with viral hepatitis [211], and its concurrent overexpression with TLR9 was correlated with a poor prognosis in HCC patients [226].

TLR4 activation, mainly mediated by DMAPs and lipopolysaccharide (LPS), its major agonist, leads to the upregulation of the NF-kβ and MAPK pathways, resulting in the secretion of pro-inflammatory mediators, such as TNFα and IL-6 [211]. The stimulated immune response plays an important role in sustaining DNA damage and supporting genomic instability that eventually leads to hepatocyte neoplastic transformation, leading to HCC development [227]. In recent years, growing evidence has suggested a more complex role for TLR4 in HCC progression, with several aspects to be considered in metastasis promotion, drug resistance, and angiogenesis support [228]. There is evidence, in HCC, of TRL4 involvement in the facilitated recruitment of T-regs to the tumor site, along with metastasis promotion through its interaction with macrophages [229]. In particular, a deviation from the canonical LPS-induced TLR4/myD88 activation and M1 polarization seems to be caused specifically by its ability to recognize distinct HCC necrotic debris, resulting in increased metastatic potential [230]. TLR4’s higher expression in relapsing hepatocellular carcinoma patients has recently been linked to its newly discovered regulatory role in the enhancement of stem properties in HCC cells via the TLR4–AKT–SOX2 pathway [231]. It has also been demonstrated that TLR4 contributes to the regulation of VEGF by LPS-activated STAT3/SP1 signaling, promoting angiogenesis [232]. To date, no clinical trial targeting TLR4 in HCC is ongoing, while the TLR4 agonist GLA-SE has been tested for the treatment of advanced soft-tissue sarcoma (NCT02180698, Phase I) and stage II-IV melanoma (NCT02320305, Early Phase I).

### 3.3. Chemokine Receptors

Chemokines are a family of small protein molecules (~10 kDa) similar in structure, function, and chemotactic properties and are divided into four subclasses: CX_3_C, CXC, CC, and XC, according to the arrangement of the two conserved cysteine residues found at their N-termini. Their receptors are grouped according to the same scheme and belong to the G-protein-coupled receptor (GPCR) family, a group of surface receptors characterized by the presence of a 7-fold transmembrane domain [233]. 

Chemokines are known to play a pivotal role in the immune system by being the major mediators of the chemotactic effect and by directly influencing the differentiation, survival, and function of immune cells [234,235,236].

It has been reported that the chemokine system has wide-ranging effects in hepatocellular carcinoma (HCC) by influencing immune and tumor microenvironment (TME) cells, leading to both anti- and pro-tumoral outcomes. As discussed below, the presence of chemokine receptors on HCC cells enables the direct modulation of the tumor cell behavior, affecting processes such as migration, proliferation, growth, and survival. Recently, a prognostic chemokine-receptor-based signature was successfully established (CCL14–CCL20/CCR3) [237], further confirming them as fertile ground to look into for finding potential therapeutic targets.

The classes of chemokine receptors mostly involved in HCC are CXCRs and CCRs.

The role of CXCL12/CXCR4–CXCR7 is well-documented, with abnormal expression levels of either CXCL12 or CXCR4/CXCR7 correlating with clinicopathological features of HCC [238,239,240,241]. The level of CXCR4 expression in HCC tissues correlates with tumor size, metastasis, and survival [242], and CXCL12/CXCR4 plays an important role in the epithelial–mesenchymal transition, with TGFβ-induced EMT correlating with high CXCR4 expression in tumor tissues, particularly corresponding to tumor borders and perivascular areas [243]. Besides TGFβ, other molecules present in the TME, such as osteopontin and astrocyte-elevated-gene-1, significantly upregulated the expression of CXCR4 in HCC [244,245]. The CXCL12/CXCR4 axis is also known to increase hypoxia-inducible factors and activate MMPs, providing another way to promote angiogenesis and cancer metastasis simultaneously [246]. All in all, CXCR4 seems to be a valuable target in HCC, and with its agonist Mozobil™ (AMD3100) already approved by the FDA [247], there is a solid possibility that its testing in HCC combined therapy will happen soon.

Other important chemokine axes are worth mentioning though. For instance, CXCL9–CXCL10/CXCR3, which has prognostic value in HCC patients [248], along with CXCL1–CXCL2/CXCR2, for which upregulation is an indicator of increased risk for HCC development [249]. The CXCL5/CXCR2 axis activates the PI3K/AKT/GSK-3/snail signaling pathway, inducing EMT in HCC cells and significantly enhancing the invasive and migratory properties of HCC cells [250]. Finally, high levels of CXCR6 found in HCC tissues result in increased expressions of IL-6 and IL-8 and correlate with higher neutrophil infiltration and worse prognoses for HCC patients, through the generation and maintenance of an inflammatory environment and promotion of tumor invasiveness [251].

In the second subclass of interest, several receptors need to be considered. Expression levels and genetic alterations of CCL2/CCR2 affect HCC patients’ prognoses [252], with CCL2 promoting angiogenesis initiation and HCC progression. This is accompanied by increased invasiveness and EMT, along with the activation of the Hedgehog pathway [253].

The CCL5/CCR5 axis is tightly connected to chronic liver inflammation and actively linked to HCC development [254]. The expression of both CCL5 and CCR5 in HCC tissues is significantly higher than in healthy liver tissues, and the CCR5 antagonist maraviroc can reverse the CCL5-mediated activation of the PI3K/AKT/mTOR pathway, inhibiting HCC progression in vivo [255]. Also, CCR10 was found to be significantly upregulated in inflammation-driven HCC tumors and in hepatocytes of paracancerous tissue, where it is secreted, activating PI3K/AKT signaling, inhibiting apoptosis, and promoting cell proliferation, leading to HCC onset [256]. Finally, CCL20/CCR6 promotes the adhesion, proliferation, and chemotactic migration of HCC cells [257]. There is evidence that blocking CCR6 activity in the tumor microenvironment inhibits tumor neovascularization and, therefore, might enhance traditional therapy’s effectiveness [258].

### 3.4. RAS/MAPK Pathway

The mitogen-activated protein kinase (MAPK) pathway is responsible for coordinating different cell functions, like cell proliferation, metabolism, migration, and differentiation, and responding to apoptosis. Its signaling cascade is extremely complex, and growth factors and mitogens use it to transmit signals from their receptors to regulate gene expression and prevent apoptosis [259]. Given its central role in controlling cell functions, the dysregulation of the MAPK pathway is associated with tumorigenesis, cancer progression, and drug resistance [260]. The pathway is triggered by the activation of proteins belonging to the RAF family (ARAF, BRAF, and CRAF), which are serine/threonine kinases that act as mediators between the upstream proteins, RAS/GTPases, and downstream kinases, such as MEK/ERK, c-Jun NH2-terminal kinase (JNK), and p38. Extracellular signaling mediated by receptors or physical stressors can activate the MAPK pathway physiologically [261]. The aberrant activation of the pathway can depend on the dysregulated activation of signal receptors (discussed previously in this review) or of upstream and downstream kinases involving mainly the MEK/ERK signal transduction pathway [262]. In cancer, the upstream RAS proteins are the proteins that are the most frequently mutated, and the RAS mutational and activation burdens can predict the prognosis, drug sensitivity, and survival. Mutations in NRAS, KRAS, and HRAS are found in 20% of cancers, with KRAS being the most frequently mutated protein (15%). RAF proteins, which are RAS upstream effectors, are also mutated in 20% of cancers, with BRAF and ARAF being the most frequently mutated (10% and 8%, respectively). Cancers have lower rates of mutations in MEK protein kinases, which are downstream mediators of RAS/RAF signaling (MAK1 2%, MAK2 5%, and ERK 3%) [263,264].

In HCC, the dysregulated activation of the MAPK pathway is the most frequently related to aberrant upstream signaling rather than gain-of-function mutations in RAS/RAF/MEK proteins. The mutational frequency of RAS proteins in HCC is 4%; in pancreatic cancer, it is 90%; and in colorectal cancer, it is 62%. However, the RAS/MAPK pathway is activated in 50–100% of human HCCs and is correlated with a poor prognosis [265,266]. CRAF and BRAF activation were confirmed to be altered in HCC [64,267], and CRAF overexpression is considered as a marker of poor prognosis in HCC [265,268]. 

Few drugs targeting MAPK pathways have been approved by the FDA and EMA, even though many molecules are being tested in preclinical models. In HCC, the MEK1 inhibitor selumetinib has already been tested without success in a Phase I clinical trial in patients with advanced cancer [269]. A similar outcome has been reported for the MEK1 and MEK2 inhibitor trametinib, which has been tested in combination with sorafenib in patients with advanced HCC (NCT02292173) [270]. 

### 3.5. JAK/STAT Pathway

The Janus kinase (JAK)/signal transducer and activator of transcription (STAT) signaling pathway is activated by various cytokines and growth factors. In humans, four members of the JAK family (e.g., JAK1, JAK2, JAK3, and TYK2) and seven members of the STAT family (e.g., STAT1, STAT2, STAT3, STAT4, STAT5A, STAT5B, and STAT6) have been described to date [271]. The activation of JAK/STAT signaling involves receptor-associated JAK activation after ligand–receptor binding, leading to the recruitment of STAT proteins to the receptor’s cytosolic domain. Phosphorylation by JAK activates STAT proteins, allowing them to form dimers, migrate into the nucleus, and bind to DNA [272]. The different combinations of JAK and STAT isoforms, along with the variety of receptors and ligands that can activate the signaling pathway make the cellular response to JAK/STAT signaling broad and, at the same time, an interesting target for drugs to block the proliferation of cancer cells [273]. The JAK/STAT signaling pathway is indeed involved in many cellular functions that are altered in cancer, such as cell differentiation, proliferation, stemness, regulation of apoptotic signaling, and modulation of the immune or inflammatory response [274,275,276,277]. In hepatocytes, JAK/STAT signaling can also modulate the metabolism by regulating gluconeogenesis [278]. In addition to the above-mentioned changes in growth-factor- and cytokine-mediated signaling, JAK/STAT signaling can be altered by activating mutations that can drive and/or maintain carcinogenesis [279,280,281,282]. A whole-genome analysis of primary HCC [283] revealed that 9.1% of patients carry activating mutations in JAK1, and a proteogenomic characterization of virus-associated liver cancers [64] showed that JAK2 is overexpressed in a subclass of HCC. Together with the Wnt pathway, JAK/STAT is the main oncogenic pathway in HCC [283]. JAK-specific inhibitors, such as tofacitinib [284], baricitinib [285], and upadacitinib [286], are currently approved for the treatment of rheumatoid arthritis and are effective in reducing cytokine-mediated inflammation [287]. Many JAK inhibitors are being tested in clinical trials for the treatment of cancer [288], mostly lymphoproliferative and myelodysplastic malignancies, and, to date, no clinical trial using JAK inhibitors has involved HCC patients [289,290,291]. There are some reports in the literature of the preclinical testing of JAK inhibitors on HCC cell lines or tumor-derived cells, generally showing a reduction in tumor cell proliferation and migration [292,293,294,295,296].

STAT3 is an emerging candidate for JAK/STAT-targeted therapy as its constitutive activation is associated with oncogenesis, metastasis, and drug resistance in various cancers, including HCC [297,298]. Nevertheless, there is no FDA- or EMA-approved drug that selectively targets STAT3. Some STAT3-specific inhibitors are currently being tested in clinical trials both as a single agent and in combination with other drugs. These include napabucasin [299,300], OPB-111077 [301,302], OPB-31121 [303,304], TT1-101 (NCT03195699), and AZD9150 [305,306], which are also being tested in Phase I or II clinical trials in HCC. Preclinical studies have demonstrated the efficacy of STAT3 inhibitor molecules in improving radiosensitivity of HCC and in reducing proliferation and promoting apoptosis [307]. It is worth noting that STAT3 is one of the targets of sorafenib, a generic kinase inhibitor approved for the treatment of unresectable HCC, and that the inhibition of STAT3 is essential for the sensitization of HCC cells to an agonistic DR5 antibody (LBY135) and for TRAIL-induced apoptosis in TRAIL-resistant HCC cells [308].

Aberrant STAT3 activation in HCC is mainly mediated by interleukin-6 (IL-6) trans-signaling and is, therefore, commonly referred to as IL-6/STAT3 axis alteration [309,310,311]. In the TME of HCC, this pro-inflammatory interleukin is generally secreted by tumor-associated macrophages, as well as tumor cells [310,312,313]. IL-6 also favors fibroblasts to acquire a pro-tumoral phenotype (CAF, cancer associated fibroblasts) [314].

Moreover, several in vitro studies support that an elevated IL-6 level in the HCC TME promotes the development of resistance to sorafenib and anti-PD-L1 antibodies [315,316,317].

There are currently no IL-6 inhibitors in clinical trials for the treatment of HCC. 

Clinically, several monoclonal antibodies have been shown to neutralize IL-6, such as siltuximab [318], sirukumab [319], and tocilizumab. The latter in particular has shown good results in the treatment of non-small-cell lung cancer [320], multiple myeloma [320], epithelial ovarian cancer [321], B-cell lymphoma [322], and renal cell carcinoma [323].

All in all, targeting the JAK/STAT metabolic pathway in HCC may be a useful therapeutic option, as it involves several oncogenic and progression-promoting mechanisms typical of HCC, such as chronic inflammation (Fibrosis is associated with oncogenesis and the development of HCC [324,325]), altered metabolism (STAT3 activation enhances the Warburg effect by promoting anaerobic glucose metabolism [326]), and angiogenesis [297,327,328].

### 3.6. PI3K/AKT/mTOR

The phosphatidylinositol-3 kinase (PI3K), protein kinase B (AKT), and mechanistic target of rapamycin (mTOR) are central control centers for many important cellular functions, such as proliferation, survival, migration, metabolism, and responses to stressors and drugs. Several upstream regulators and downstream effectors are involved in this complex signaling axis, each of which may play a role in carcinogenesis. PI3K belongs to a family of lipid kinases that are categorized into three groups based on their substrate specificity, structure, and mechanism (see [329] for a more detailed description of PI3K proteins). The activation of PI3K triggers the conversion from phosphatidylinositol-4,5-biphosphate (PIP2) to phosphatidylinositol-3,4,5-triphosphate (PIP3), which serves as a secondary messenger to initiate the activation, by phosphorylation, of AKT mediated by PDK-1 or mTORC2. This process is highly tuned and can be antagonized by various phosphatases, including PTEN, a known tumor suppressor. Activated AKT phosphorylates and activates multiple downstream targets, including the RHEB protein, which is released and can activate mTOR. The activated mTOR protein can organize itself into two complexes: mTORC1 and mTORC2, the latter of which is exclusively dependent on PIP3 phosphorylation for its activation. PI3K-mediated mTORC2 activation can activate mTORC1 via AKT. A more detailed overview of PI3K/AKT/mTOR signaling can be found in [330,331].

Seventeen genes are involved in controlling the PI3A/AKT/mTOR signaling pathway: *PIK3CA*, *PIK3R1*, *PIK3R2*, *PTEN*, *PDPK1*, *AKT1*, *AKT2*, *FOXO1*, *FOXO3*, *MTOR*, *RICTOR*, *TSC1*, *TSC2*, *RHEB*, *AKT1S1*, *RPTOR*, and *MLST8*. In cancer, *PIK3CA* is most frequently mutated, with mutations in AKT1, AKT2, and the proteins of the mTOR complex, except for RICTOR and mTOR, being relatively uncommon [332].

The PI3K/AKT/mTOR signaling pathway is altered in 51% of HCCs [333], with the most frequently mutated genes being *PTEN* (5%), *PIK3CA* (4%), *MTOR* (4%), and *AKT2* (2%) [263,333]. According to Fujita et al. [64], *SYK* is selectively upregulated in the R1 proteomic class, *RHEB* is selectively upregulated in the R2 proteomic class, and *AKT1*, *LK1*, *pS6K*, and *DUSP4* are selectively upregulated in the R3 proteomic class.

FDA-approved or investigational inhibitors of the PI3K/AKT/mTOR pathway are being tested for the treatment of HCC either in preclinical or clinical trials. Copanlisib, a PI3K inhibitor, is being tested in a clinical trial in combination with nivolumab (NCT03735628) and has shown potent antiproliferative activity in an in vitro model in combination with sorafenib [334]. MK2206, an AKT inhibitor, is being tested in HCC patients who have not responded to prior therapy (NCT01239355). Everolimus, an mTOR inhibitor, is used alone [335] or in combination with TACE (NCT01239355) or sorafenib [336] in clinical trials in HCC patients. A dual mTORC1/mTORC2 inhibitor, CC-223, is being tested in HBV-positive HCC patients (NCT03591965). Sapanisertib, an mTOR inhibitor, is being tested in advanced or metastatic HCC patients (NCT02575339). A more detailed analysis of clinical trials targeting PI3K/AKT/mTOR in HCC can be found in [337].

To date, targeting PI3K/AKT/mTOR in HCC has yielded only partially positive results, mainly due to early onset drug resistance and cancer progression [337].

### 3.7. Wnt/β-Catenin

Wnt/β-catenin is a complex pathway that regulates several cellular functions. Canonical Wnt signaling is repressed in adult tissues by the intervention of several antagonists and inhibitors. Wnt1 and Wnt3a, ligands activating Wnt/β-catenin signaling, belong to a family of 19 proteins. Their activation requires lipid modification mediated by the porcupine O-acyltransferase (PORCN) protein. β-catenin, a structural protein in cell adhesion, is encoded by the *CTNNB1* gene. In the absence of Wnt signaling, *CTNNB1* is poorly expressed, and its translocation to the nucleus is impaired by a protein complex named the destruction complex, maintaining low β-catenin levels through proteasomal degradation.

Upon Wnt signaling activation, β-catenin becomes free from the scaffold protein AXIN and is released from the destruction complex. Cytosolic β-catenin can translocate to the nucleus and act as a scaffold for the formation of transcriptional factor complexes, activating the expression of downstream effectors of the Wnt/β-catenin signaling. A comprehensive description of the signaling pathway can be found in [338].

According to TCGA data, twenty-five genes are involved in Wnt signaling tumorigenesis and cancer progression. These include SFRP1, SFRP2, SFRP4, SFRP5, SOST, TCF7L1, TLE1, TLE2, TLE3, TLE4, WIF1, ZNRF3, CTNNB1, AMER1, APC, AXIN1, AXIN2, DKK1, DKK2, DKK3, DKK4, GSK3B, RNF43, TCF7, and TCF7L2 [263].

In hepatic cells, Wnt/β-catenin is activated under physiological conditions and is essential for hepatobiliary development and regeneration processes [339]. It is one of the most dysregulated pathways in HCC, with mutations detected in 62.5% of HCC samples [283]. The most frequent mutations in HCC involve *CTNNB1* (15.9–30%), *AXIN1* (4.5–8%), and *APC* (0.8–3%) [263,283,340]. DKK1 is considered as an early-stage HCC biomarker [341,342], and CTNNB1 is enriched in proteomic subclass 3 [64]. CTNNB1 mutations are significantly associated with telomerase reverse-transcriptase promoter (TERT) mutations [343]. The telomerase proteins are responsible for the maintenance of chromosomal and genomic integrity; correlation between CTNNB1 and TERT mutations in HCC may indicate a connection between CTNNB1 mutations and genomic instability [343,344].

Despite several Wnt/β-catenin inhibitors being selected and proven effective in blocking cancer proliferation in preclinical models, very few have reached the clinical trial stage, and none have gained FDA or EMA approval for cancer treatment [345]. Among the drugs tested in clinical trials, E7386, the first-in-class orally active β-catenin antagonist, is being tested in HCC in combination with lenvatinib (NCT04008797) or pembrolizumab (NCT05091346) [346]. CGX1321, a PORCN inhibitor, and DKN-01-mAb, a DKK1 inhibitor, are undergoing clinical trials in combination with other drugs in patients with advanced liver cancers (NCT02675946 and NCT03645980, respectively). OMP-54F28, an antagonist of the FZD8/Wnt complex, has been tested in Phase I clinical trials for the treatment of HCC (NCT02069145).

A comprehensive review of preclinical studies targeting Wnt/β-catenin can be found in [341,342].

### 3.8. p53

The protein p53 is a transcription factor and regulates the expression of genes involved in the cell cycle’s arrest, apoptosis, and DNA repair [347]. In addition to its primary functions, p53 also plays an important role in other cellular and metabolic processes. A more detailed analysis of the functions of p53 can be found in [348]. Given its important role in vital cellular processes, the expression and function of p53 are highly regulated [349,350]. Under physiological and homeostatic conditions, the expression levels of p53 are very low, and its ability to translocate to the nucleus is regulated by various inhibitors. Among others, MDM2, an E3 ubiquitin ligase localized in the cell nucleus, plays a crucial role in maintaining low p53 levels [351].

The oncosuppressor gene *TP53* is one of the most frequently mutated genes in cancer. In fact, somatic mutations in *TP53* are found in about 34–50% of samples from cancer patients [263,352]. Mutations in *TP53* are considered as driver mutations that lead to malignant transformation [353].

Although 70% of all *TP53* alterations are missense mutations [354], alterations in the p53 signaling pathway may also depend on mutations in proteins involved in the negative feedback regulation of the p53 activity [355]. 

p53 is considered as a druggable target, and the main targeting strategies include the use of small molecules that induce the refolding of the mutant p53 and, thus, restore its pro-apoptotic activity; small-molecule inhibitors of MDM2–p53 interactions that block the aberrant degradation of p53 also in the presence of the wild-type p53 protein; and gene therapy aimed at reestablishing the normal levels and function of p53. A complete review of p53-targeting strategies can be found in [354,356].

In HCC, *TP53* is mutated in 30.8–35.2% of patients [283,357,358], and *TP53* mutations are significantly associated with reduced protein levels [64] and a poor prognosis [359]. To date, there are no specific clinical trials targeting p53 in HCC, but several molecules are being tested for the treatment of solid tumors: ASTX295 (NCT03975387), HDM201 (NCT04116541), idasanutlin MT (NCT04589845), and milademetan MT (NCT0512397), which are small molecules that inhibit MDM2’s activity; PC14586 (NCT04585750) and arsenic trioxide MT (NCT04869475), two small molecules that target mutant p53; and Ad-p53, a p53-based gene therapy (NCT03544723). Ad-p53, a recombinant human adenovirus p53, was approved by the China Food and Drug Administration (CFDA) in 2003 as the first gene therapy for the treatment of cancer [360,361]. A recent systematic review of 17 clinical trials in which Ad-p53 was used in combination with TACE for the treatment of HCC found that the addition of recombinant p53 resulted in improved overall survival and quality of life compared to standard monoagent TACE therapy [362]. Another study using Ad-p53 in combination with fractionated stereotactic radiotherapy showed improvement in the outcome of the combined therapy compared to fractionated stereotactic radiotherapy alone [363].

Given its prominent role in malignant transformation and cancer progression and its high mutation frequency in cancers [352], p53 is an attractive target in cancer treatment, as evidenced by the increasing number of studies on potential therapeutic approaches to restore the pro-apoptotic function of p53 [354,356,364,365]. Nevertheless, no p53-specific drug has yet received FDA or EMA approval for the treatment of cancer [354,365]. 

### 3.9. Cyclins and Cyclin-Dependent Kinases

The cell cycle’s progression is promoted, maintained, and regulated by the interplay of cyclins and specific cyclin-dependent kinases (CDKs), which, in turn, are controlled by a variety of extracellular and intracellular signaling [366]. CDKs are a family of 20 small proteins with serine–threonine kinase activity for which activation depends on their interaction with cyclins, a class of proteins for which the concentration varies greatly during each phase of the cell cycle [367]. CDK2, CDK4, and CDK6 are cell-cycle-related CDKs, while the others are involved in transcriptional regulation, the maintenance of genomic stability, and other biological processes [368]. The activation of the CDK–cyclin complex is tightly regulated and depends on the activity of specific phosphatases and CDK-activating kinases (CAKs) [366].

Given their role in controlling the cell cycle’s progression, the dysregulation of CDKs and cyclins is a hallmark of cancer and, therefore, of great interest as targets for cancer treatment [369]. In particular, CDK4 and CDK6, as well as cyclins D1 and E, are frequently altered in cancer [369].

According to whole-genome and exome sequencing data, *CCND1* and *CCNE1*, encoding cyclins D1 and E1, respectively, are amplified in 4.5% of HCC patients, while *CDKN2A* and *CDKN2B*, two negative regulators of CDKs, are deleted in 10.2% and 12.5% of patients, respectively [283,333]. Moreover, the overexpression of a set of cell cycle proteins comprising CDK1 and cyclin B1 was significantly associated with poor overall survival in HCC patients [64]. By proteomic profiling, WEE1, a negative regulator of the CDK1–cyclin B complex, was overexpressed in the R3 subclass comprising HCCs that were smaller in size and less invasive, but with poor immune infiltration [64].

CDKs are the primary target for cancer therapy against the cell cycle. The first molecules in this class were pan-CDK inhibitors (CDKi) [370,371], but their use was very limited and discontinued owing to their high toxicity [372,373]. Second-generation CDK inhibitors are much more selective for CDK4/6 and have fewer side effects [374,375]. Three CDKi molecules are approved by the EMA and FDA for the treatment of HR-positive, HER2-negative breast cancer: palbociclib [376], ribociclib [377], and abemaciclib [378]. In HCC, CDKi molecules have an antiproliferative activity when tested in preclinical models [379,380,381,382,383] and are being tested in clinical trials either alone or in combination with other therapeutics (NCT01356628, NCT02524119, and NCT03781960), but the results are not yet available.

Targeting cell cycle regulators, such as CDKs, can be a valuable strategy to stop cancer cell proliferation and induce cell death, as proved by the improvement in the treatment of breast cancer using CDK4/6 inhibitors. A similar strategy can result in improved treatment of HCC subtypes in which cell cycle deregulation depends mostly on CDK/cyclin deregulation. 

### 3.10. TGFβ Signaling

The TME is a complex environment that is grafted around the tumor, in which intricate interactions are established between cellular (e.g., tumor-associated macrophages (TAMs), tumor-associated neutrophils (TANs), and myeloid-derived suppressor cells (MDSCs)) and noncellular (e.g., extracellular matrix (ECM) proteins, proteolytic enzymes, cytokines, and growth factors) elements, which cooperate to favor tumor growth and progression.

The need to identify therapeutic strategies that target the TME is becoming increasingly urgent given the fundamental role it plays in the development of drug resistance mechanisms and tumorigenesis. The TME of HCC is particularly hostile to immunotherapy, creating an immunosuppressive environment that easily undermines the efficacy of this type of therapeutic approach. Given the crucial role of TFGβ in the TME crosstalk, this section will focus on this signaling as a strong candidate target. A more detailed review analyzing other potential targets in the TME of HCC can be found in [206].

The TFGβ superfamily consists of 33 multifunctional cytokines, of which TGFβ1 represents the most abundant isoform. TFGβ signaling is critically involved in the regulation of multiple cellular processes controlling tissue homeostasis, including proliferation, differentiation, migration, and cell death. TFGβ signaling dysregulation is, therefore, detected in many diseases, including hepatocarcinoma [384,385,386,387].

TGFβ has two opposing roles in HCC development. In the early stages of the disease, it plays an inhibitory role in tumor growth by inducing apoptosis in cancer cells [388,389,390]. However, along the process, tumor cells become resistant to its inhibitory action, owing to the inactivation of the key components in the pathway or the hyperactivation of parallel pathways that counteract its inhibitory action. At this point, tumor cells begin to respond to its presence by inducing EMT, which increases their migratory capabilities and invasive potential [391,392]. In this process, the growth factors EGF and PDGF play an important role by activating an autocrine loop that increases the ability of the cells to overcome the pro-apoptotic effect of TFGβ [92,393]. Recent evidence has also been shown of a possible involvement of TFGβ in the metabolic reprogramming of tumor cells [394]. 

TFGβ plays an equally important role as a mediator among TME components throughout the cancer development process.

Among others, HSC-derived TGFβ is known to upregulate NNMT (N-methyltransferase) expression in HCC [395]. NNMT correlates with tumor stages and DFS in HCC, indicating its prognostic significance. Therefore, potential therapeutic strategies targeting NNMT in HCC may be worth exploring, including NNMT inhibitors and drug repurposing with statins, although further clinical studies are needed [396,397]. 

The idea for using TFGβ as a therapeutic target is becoming increasingly attractive. However, it is necessary to correctly identify patients where TFGβ is exerting a pro-tumoral action and who could, therefore, benefit from such a therapeutic approach. Several studies conducted in vitro, in vivo, and in HCC-patient-derived material allowed the identification of the so-called “late TFGβ signature”, which is the selective overexpressions of CXCR4, CD44, SMAD7, or CTLC genes in concurrence with the TFGβ one. This could be of great help in the identification of patients who could benefit from an anti-TFGβ treatment [243,392,398,399,400,401]. 

Several TFG-β inhibitors are available and have different mechanisms of action. Some suppress the production of TFGβ itself, others inhibit its activity, and others block either its interaction with the receptor or the kinase activity of the receptor itself. These inhibitors have shown promising anti-tumor activity in several in vivo studies, and some are also being tested in clinical trials in patients with advanced HCC (NCT01246986, NCT02906397, NCT02423343, and NCT02699515) [387,402,403].

Recently, a CAR-T called PSMA–TGFβRDN has also been developed to neutralize TGFβ’s pro-tumoral activity. T-cells engineered with this CAR have shown increased proliferative capacity and persistence in vivo, as well as increased cytokine release and reduced exhaustion. CAR–PSMA–TGFβRDN is being tested in Phase I clinical trials in patients with advanced or relapsed prostate cancer, though no clinical trial has been planned yet in patients with HCC (NCT03089203 and NCT04227275) [404].

## 4. Conclusions

The identification of effective therapeutic targets in HCC remains an open challenge in cancer research. The complexity of the tumor ecosystem, along with its distinctive immunosuppressive microenvironment, has made the development of effective targeting strategies difficult to achieve. Although data collection on the mRNA expression levels from tumor samples remains a powerful tool, it is not exhaustive when searching for new potential targets. In fact, during the writing of this review, it became increasingly clear that in the search for new targets, there is a need for an in-depth analysis at the level of protein expression as well as of the activation status of the proteins involved in the major signaling pathways described above. 

The roles of most of them and their potential as targets for new therapeutic options for HCC have been examined in this review. However, there are many other valid candidates that need to be thoroughly examined for their roles in HCC tumorigenesis, cancer progression, and prognosis. 

In conclusion, in this review, we have discussed the currently available approaches to target this difficult-to-treat cancer and summarized potential emerging protein targets, summarized in Table 2 and Table 3, with the understanding that much work remains to be done to gain a clearer view of the global situation to establish more effective targeting strategies.

**Table 2 cancers-16-00901-t002:** Molecules targeted in Phase III trials.

Target Class	Molecule	References	Clinical Trial IDs
VEGF/VEGFRs	VEGFR2	[64,164,165]	NCT02435433 (Phase III) NCT01140347 (Phase III)
	VEGF	[64,157,162,163,164,166,167,168]	NCT04487067 (Phase III) NCT04732286 (Phase III) NCT04102098 (Phase III) NCT03434379 (Phase III) NCT05904886 (Phase III)
HGF/c-Met	c-MET	[24,187,194,195,196]	NCT01755767 (Phase III) NCT01908426 (Phase III) NCT03755791 (Phase III)

**Table 3 cancers-16-00901-t003:** Molecules targeted in pre-clinical and Phase I/II trials.

Target Class	Molecule	References	Clinical Trial IDs
EGF/EGFRs	EGFR	[66,72,74,75,76,77,78]	
	ERRfI1	[64,83]	
	TGFα	[70,90,91]	
	EGF	[70,92]	
	ADAM17	[70]	
	ERBB2	[93]	
	ERBB3	[93,94]	
	NRG1	[95]	
PDGF/PDGFRs	PDGFRα	[93,116,118,123,125,126]	
	PDGFRβ	[93,117,123,126]	
FGF/FGFRs	FGFR3	[129,130]	
	FGFR4	[93,131,140,142,146]	NCT04194801 (Phase I/II) NCT02508467 (Phase I)
	FGF1	[132,133]	
	FGF2	[136,137,138]	
	FGF8	[128,139]	
	FGF19	[140,142,143,144,148,149,150]	
VEGF/VEGFRs	VEGFR1	[64,164,165]	
HGF/c-Met	c-MET	[24,187,194,195,196]	NCT01988493 (Phase I/II) NCT01737827 (Phase II) NCT03672305 (Phase I)
	HGF	[188,193]	
Toll-like Receptors	TLR2	[219,220,221,222]	NCT05937295 (Phase I)
	TLR4	[211,226,227,228,229,230,231,263]	
Chemokine Receptors and Ligands	CXCL12/CXCR7 axis	[239,240]	
	CXCL12/CXCR4 axis	[238,241,242,243,244,245]	
	CXCL9–CXCL10/CXCR3 axis	[248]	
	CXCL1–CXCL2/ CXCR2 axis	[249]	
	CXCL5/CXCR2 axis	[250]	
	CXCR6	[251]	
	CCL2/CCR2	[252,253]	
	CCL5/CCR5	[254,255]	
	CCL20/CCR6	[257,258]	
	CCR10	[256]	
RAS/MAPK	CRAF	[64,267,268]	
	BRAF	[64,267]	
	MEK1 and MEK2	[266,269]	NCT00604721 (Phase II) NCT02292173 (Phase I)
JAK/STAT	JAK1	[283,296]	
	JAK2	[64,292,294,295]	
	STAT3	[297,298,299,301,307,308,313,317]	NCT03195699 (Phase I)
	IL-6	[309,310,311,312,314,315,316]	
PI3K/AKT/mTOR	PTEN	[263,333]	
	PI3K	[263,333,334]	NCT03735628 (Phase I)
	SYK	[64]	
	RHEB	[64]	
	AKT	[64,263,333]	NCT01239355 (Phase II)
	mTOR	[263,333,335,336]	NCT01239355 (Phase II) NCT03591965 (Phase II) NCT02575339 (Phase I/II)
Wnt/β-catenin	β-catenin	[263,283,340]	NCT04008797 (Phase I) NCT05091346 (Phase I/II)
	APC	[64,263,283,340]	
	AXIN1/AXIN2	[263,283,340]	
	DKK1	[341,342]	NCT03645980 (Phase I/II)
	TERT	[343,344]	
	PORCN	[283]	NCT02675946 (Phase I)
	FZD8/Wnt complex	[283]	NCT02069145 (Phase I)
P53 and Cell Cycle	P53	[64,283,317,357,358,359,362,381]	
	Cyclin D1	[283,333]	
	Cyclin E1	[283,333]	
	CDKN2A, CDKN2B	[283,333]	
	CDK1	[64]	
	WEE1	[64]	
	CDK4/6	[379,380,381,382,383]	NCT01356628 (Phase II) NCT02524119 (Phase II) NCT03781960 (Phase II)
Tumor Microenvironment	TGFβ	[243,387,388,391,392,398,399,401,402,403]	NCT01246986 (Phase II) NCT02906397 (Phase I) NCT02423343 (Phase I/II) NCT02699515 (Phase I)

## Figures and Tables

**Figure 1 cancers-16-00901-f001:**
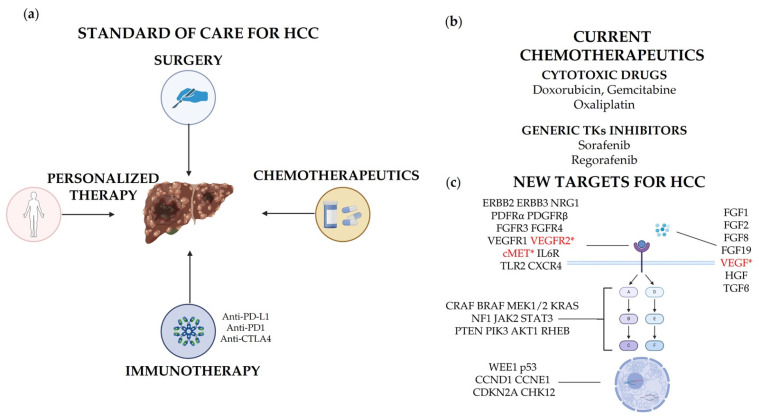
(**a**) Schematic representation of therapeutic options for the treatment of hepatocellular carcinoma (HCC). (**b**) Drugs currently approved for the treatment of unresectable HCC. (**c**) Potential new targets for personalized HCC treatment, as selected in the present review. Targets with drugs in Phase III trials are indicated by an asterisk and red. Refer to Tables for details and current clinical trials targeting the selected ligands, receptors, and pathways.

**Table 1 cancers-16-00901-t001:** Antibodies used in HCC therapy. N.A.: not applicable.

Drug	Commercial Name	Target	Format	Reference	Drug Associated in Combination Therapy
Atezolizumab	Tecentriq™	PD-L1 (CD274)	Fc optimized (no ADCC or CDC), humanized	[18]	Bevacizumab
Durvalumab	Imfinzi™	PD-L1	Monoclonal	[19]	Tremelimumab
Pembrolizumab	Keytruda®	PD-1	Humanized, IgG4 Ab	[20]	N.A.
Nivolumab	Opdivo™	PD-1	Human, IgG4 Ab	[21]	Ipilimumab
Ipilimumab	Yervoy™	CTLA4	Human, IgG1 Ab	[21]	Nivolumab
Tremelimumab	Imjudo®	CTLA4	Human, IgG2 Ab	[19]	Durvalumab

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
