# Peer review of "Hepatocellular Carcinoma: Old and Emerging Therapeutic Targets"

_cancers, 2024, doi:10.3390/cancers16050901_

Round 1

Reviewer 1 Report

Comments and Suggestions for Authors

The present Review is a very long one and not well organized. The authors should focus on the currently used therapeutic strategies (with a Table with all Phase III registration Trials+Figure showing the most important targets). The second part of the manuscript should focus only to the promising new strategies again with a Table with initial clinical results plus a Figure with their novel mechanism of action.

Author Response

We thank the Reviewer for his constructive comments.

In the text, we had already provided an early paragraph focused on the state of the art of HCC treatment and of targeted therapy (page 2), followed by the core of the content. In Section 3, in fact, we discussed current and emerging targets, also including, where appropriate, their clinical stage of development.

While we understand the Reviewer's point of view, for the discussion of such targets we decided to adopt a structure centered on the main molecular alterations found in the cancer biology of HCC according to recent “omic” work, rather than going deep into the state of development of already established therapeutic strategies. We felt this to be more appropriate to effectively help the readers to direct their search for new targets, even if they are still at a preliminary level of investigation.

The figure reporting the targets was integrated, as per the Reviewer’s indication, with an asterisk and red color to indicate targets with drugs in Phase III. Finally, a separate Table (Table 2) was built with this information in more extended form, separating the Phase III targets from those in Phase I/II (Table 3).

We honestly found the mechanisms of action too wide in their variety to be represented in a single figure and opted for maintaining their description only in the text.

Reviewer 2 Report

Comments and Suggestions for Authors

The review article “Hepatocellular Carcinoma: old and emerging therapeutic targets” by Pessino et. al. provides a detailed overview of currently existing and novel targets of high therapeutic relevance. The narrative review covers a wide variety of emerging molecular targets that are of significant clinical importance in treating Liver cancer, one of the trickiest solid tumors to treat. The review provides a detailed list of key molecules in pre-clinical studies or in clinical trials being studied for HCC targeting.

The review provides comprehensive understanding in the field of molecular targeting in Hepatocellular Carcinoma. The review also adds broader clinical understanding behind the currently available approaches for HCC treatment and emphasizes on the need to develop more effective strategies for HCC targeting. The review can be accepted in current format.

Author Response

We thank the Reviewer for their comment. We are glad to see that the Reviewer appreciated how the work was structured and developed.

Reviewer 3 Report

Comments and Suggestions for Authors

Pessino et al presented a review on Hepatocellular Carcinoma focusing on current and advanced approaches to treat HCC, considering both known and new potential targets. Authors collected recent reports on antibody drug conjugates (ADCs), chimeric antigen 27 receptor T-cell therapy (CAR-T) and engineered antibodies. Following minor corrections need to be implemented:-

1. In intro part, the cancer statistics presented as per year 2020, this needs to be updated 

2. References are seems to be around year 2020, authors should focus recent 3 years in more details.

3. Grammatical and typo errors must be corrected throughout the manuscript

Author Response

  1. In intro part, the cancer statistics presented as per year 2020, this needs to be updated 

 We thank the Reviewer for their constructive comment. We addressed their suggestions as follows:

To the best of our knowledge, the most recent reference studies for cancer incidence statistics are the Global Burden of Disease 2019 and GLOBOCAN 2020, as reported in several works published last year:

Oh JH, Jun DW. The latest global burden of liver cancer: A past and present threat. Clin Mol Hepatol. 2023 Apr;29(2):355-357. doi: 10.3350/cmh.2023.0070. Epub 2023 Mar 9. PMID: 36891606; PMCID: PMC10121295.

Choi S, Kim BK, Yon DK, Lee SW, Lee HG, Chang HH, Park S, Koyanagi A, Jacob L, Dragioti E, Radua J, Shin JI, Kim SU, Smith L. Global burden of primary liver cancer and its association with underlying aetiologies, sociodemographic status, and sex differences from 1990-2019: A DALY-based analysis of the Global Burden of Disease 2019 study. Clin Mol Hepatol. 2023 Apr;29(2):433-452. doi: 10.3350/cmh.2022.0316. Epub 2023 Jan 4. PMID: 36597018; PMCID: PMC10121317.

Rumgay H, Arnold M, Ferlay J, Lesi O, Cabasag CJ, Vignat J, Laversanne M, McGlynn KA, Soerjomataram I. Global burden of primary liver cancer in 2020 and predictions to 2040. J Hepatol. 2022 Dec;77(6):1598-1606. doi: 10.1016/j.jhep.2022.08.021. Epub 2022 Oct 5. PMID: 36208844; PMCID: PMC9670241. 

More recent updates of these databases are not yet available and published.

  1. References are seems to be around year 2020, authors should focus recent 3 years in more details.

A careful and extensive review of the literature on the subject, also including the relevant publications of the last three years, has already been provide in the text. The very broad nature of the subject of the review, which has long been of interest, precisely because of the difficulty in finding effective targeting strategies, results in a prevalence of slightly older references needed to best frame and organize the topic. Nevertheless, references to the most recent publications can be found in each paragraph.

  1. Grammatical and typo errors must be corrected throughout the manuscript

We thank the Reviewer for noticing errors and typo that we missed, and that we have corrected in this second version of the manuscript (highlighted in yellow).

Reviewer 4 Report

Comments and Suggestions for Authors

The manuscript “Hepatocellular Carcinoma: old and emerging therapeutic targets” is a review article regarding current and advanced approaches to treat HCC, considering both known and novel potential targets.

The manuscript is well written and is of interest for the readers. I really appreciate the work performed by authors.

I suggest the following modifications of the manuscript in order to improve it:

1.     I would remove the commercial name of the antibodies from the table 1.

2.     In figure 1 there is a mistake, “reforrafenib” should be “regorafenib”.

3.     The manuscript should include another target in the novel potential targets. The review ignores the enzyme nicotinamide N-methyltransferase, a phase-II enzyme which catalyzes the N-methylation of nicotinamide (PMID: 36829935). Indeed, NNMT has been found to contribute to HCC progression and aggressiveness (PMID: 19216803; PMID: 31294922). This is particularly interesting since its activity has been linked to the TGF-beta signalling pathway both in neoplasms and non-neoplastic disorders (PMID: 19216803; PMID: 19216803). Many NNMT inhibitors have been already developed, among the best should be cited Alkene-Linked Bisubstrate NNMT inhibitors, Macrocyclic peptides as allosteric inhibitors and Bisubstrate Inhibitor in the form of Esterase-Sensitive Prodrug, which could be tested for HCC treatment.

Comments on the Quality of English Language

English is fine/Minor editing required.

Author Response

We thank the Reviewer for their comment and appreciation of our work. We have addressed their criticism as follows:

  1. I would remove the commercial name of the antibodies from the table 1.

We thank the Reviewer for their suggestion. Nevertheless, we believe it beneficial to keep the commercial name of the antibodies in the tables for more immediate identification by clinicians.

  1. In figure 1 there is a mistake, “reforrafenib” should be “regorafenib”.

We thank the Reviewer for noticing the typo present in Figure 1. It was corrected as indicated. 

  1. The manuscript should include another target in the novel potential targets. The review ignores the enzyme nicotinamide N-methyltransferase, a phase-II enzyme which catalyzes the N-methylation of nicotinamide (PMID: 36829935). Indeed, NNMT has been found to contribute to HCC progression and aggressiveness (PMID: 19216803; PMID: 31294922). This is particularly interesting since its activity has been linked to the TGF-beta signalling pathway both in neoplasms and non-neoplastic disorders (PMID: 19216803; PMID: 19216803). Many NNMT inhibitors have been already developed, among the best should be cited Alkene-Linked Bisubstrate NNMT inhibitors, Macrocyclic peptides as allosteric inhibitors and Bisubstrate Inhibitor in the form of Esterase-Sensitive Prodrug, which could be tested for HCC treatment.

We thank the Reviewer for their suggestion.

The manuscript was modified as follows (line 977-982):

“Among others, HSC-derived TGFβ is known to upregulate NNMT (N-methyltransferase) expression in HCC [398]. NNMT correlates with tumor stage and DFS in HCC, indicating its prognostic significance. Therefore, potential therapeutic strategies targeting NNMT in HCC may be worth exploring, including NNMT inhibitors and drug repurposing with statins, although further clinical studies are needed [399,400].”

References

  1. Kim, J.; Hong, S.J.; Lim, E.K.; Yu, Y.S.; Kim, S.W.; Roh, J.H.; Do, I.G.; Joh, J.W.; Kim, D.S. Expression of Nicotinamide N-Methyltransferase in Hepatocellular Carcinoma Is Associated with Poor Prognosis. J. Exp. Clin. Cancer Res. 2009, 28, 20, doi:10.1186/1756-9966-28-20.
  2. Campagna, R.; Vignini, A. NAD+ Homeostasis and NAD+-Consuming Enzymes: Implications for Vascular Health. Antioxidants (Basel, Switzerland) 2023, 12, doi:10.3390/ANTIOX12020376.
  3. Li, J.; You, S.; Zhang, S.; Hu, Q.; Wang, F.; Chi, X.; Zhao, W.; Xie, C.; Zhang, C.; Yu, Y.; et al. Elevated N-Methyltransferase Expression Induced by Hepatic Stellate Cells Contributes to the Metastasis of Hepatocellular Carcinoma via Regulation of the CD44v3 Isoform. Mol. Oncol. 2019, 13, 1993–2009, doi:10.1002/1878-0261.12544.

Round 2

Reviewer 1 Report

Comments and Suggestions for Authors

I think the authors adequately adressed all the comments

Comments on the Quality of English Language

I think the authors adequately adressed all the comments

Reviewer 4 Report

Comments and Suggestions for Authors

The manuscript can be published.

Comments on the Quality of English Language

Minor editing/typos required.